Limited effect of a highway barrier on the genetic structure of a gypsum soil specialist

Martín-Rodríguez Irene irene.martin@urjc.es
Escudero Adrián
http://orcid.org/0000-0002-7672-9872 García-Fernández Alfredo
Área de Biodiversidad y Conservación. Departamento de Biología y Geología, Física y Química Orgánica, Universidad Rey Juan Carlos , Móstoles, Madrid , Spain
Culham Alastair
Electronic publication date: 2021 Jan 4
Publication date: 2021
Volume: 9
Electronic Location ID: e10533
Received 2020 Jun 25; Accepted 2020 Nov 19
Copyright: © 2021 Martín-Rodríguez et al.
Copyright year: 2021
Copyright holder: Martín-Rodríguez et al.
License: This is an open access article distributed under the terms of the Creative Commons Attribution License, which permits unrestricted use, distribution, reproduction and adaptation in any medium and for any purpose provided that it is properly attributed. For attribution, the original author(s), title, publication source (PeerJ) and either DOI or URL of the article must be cited.
License URL: https://creativecommons.org/licenses/by/4.0/

Keywords: Fragmentation, Genetic marker, Gypsophile, Gypsum outcrops, Highway, Landscape genetics

Funding: ECONECT—Obrascón Huarte Lain (OHL) and Centro para el Desarrollo Tecnológico Industrial (CDTI) REMEDINAL-TE CM and P2018/EMT-4338 PHENTYPES PGC2018-099115-B-I00 This study has been financially supported by ECONECT (funded by Obrascón Huarte Lain (OHL) and Centro para el Desarrollo Tecnológico Industrial (CDTI)), (REMEDINAL-TE CM, P2018/EMT-4338) and PHENTYPES (PGC2018-099115-B-I00) projects. There was no additional external funding received for this study. The funders had no role in study design, data collection and analysis, decision to publish, or preparation of the manuscript.

==============================
Background

Gypsum ecosystems are edaphic islands surrounded by a matrix that is inhospitable to gypsum soil plant specialists. These naturally fragmented landscapes are currently exacerbated due to man-made disturbances, jeopardising their valuable biodiversity. Concomitant action of other fragmentation drivers such as linear infrastructures may increase the already high threat to these specialists. Although some evidence suggest that gypsophytes are not evolutionary dead-ends and can respond to fragmentation by means of phenotypic plasticity, the simultaneous action of barriers to genetic flow can pose a severe hazard to their viability. Here, we evaluated the effect of a highway with heavy traffic on the genetic flow and diversity in the species Lepidium subulatum, a dominant Iberian shrubby gypsophyte.

Methods

We tested the possible existence of bottlenecks, and estimated the genetic diversity, gene flow and genetic structure in the remnant populations, exploring in detail the effect of a highway as a possible barrier.

Results

Results showed variability in genetic diversity, migrants and structure. The highway had a low impact on the species since populations can retain high levels of genetic diversity and genetic parameter, like FST and FIS, did not seem to be affected. The presence of some level of genetic flow in both sides along the highway could explain the relatively high genetic diversity in the habitat remnants.

Discussion

Natural fragmentation and their exacerbation by agriculture and linear infrastructures seem to be negligible for this species and do not limit its viability. The biological features, demographic dynamics and population structures of gypsum species seem to be a valuable, adaptive pre-requisite to be a soil specialist and to maintain its competitiveness with other species in such adverse stressful conditions.

Introduction

Fragmentation is known as a pervasive anthropogenic process in which habitat loss triggers the division of the available habitat into patches, creating small, isolated and low quality fragments which are subject to the action of other concomitant negative biological processes (e.g., edge effects, decreased abundance and richness and increased mortality rate, among others, Boulinier et al., 2001; Fahrig, 2002; 2003). Connectivity is a critical parameter for long-term demographic viability in fragmented landscapes, conveying information about the species’ movement and genetic flow between suitable remnants (Nogués & Cabarga-Varona, 2014). It is usually measured by indirect surrogates, such as the number of habitat patches in a given area, the patch size or the nearest neighbour distance. In addition, some estimates of plant and population performance are usually considered when fragmentation effects want to be isolated (Fahrig, 2003; Zhou et al., 2016). Although only a minimal set of studies have used molecular markers to evaluate the connection between fragments (Martínez-Nieto et al., 2012; Gómez-Fernández, Alcocer & Matesanz, 2016), there is no doubt that fragmentation profoundly affects the gene flow between them and, also, the performance and viability of plant populations within each habitat remnant (see Fahrig, 2003). Many studies have shown that the population’s permanence and viability on one of these islands depends on the fragmentation process itself and, moreover, on dispersal ability which is critical for connecting patches of suitable habitat (Bascompte, Possingham & Roughgarden, 2002).

Linear infrastructure is one of the main causes of habitat fragmentation (Nogués & Cabarga-Varona, 2014), especially in high-income countries. Aside from the negative effects of fragmentation (Benítez-López, Alkemade & Verweij, 2010; Karlson, Mörtberg & Balfors, 2014), linear infrastructure can also trigger an additional barrier effect, which can modify genetic flow among populations (Ament et al., 2008), increasing invasive species (Forman & Alexander, 1998; Boarman & Sazaki, 2006) or increasing human access (Benítez-López, Alkemade & Verweij, 2010; Clauzel et al., 2015). On the other hand, some species are favoured by infrastructure (Karlson, Mörtberg & Balfors, 2014) because they are able to colonise new habitats created by the infrastructure, such as embankments (Forman & Alexander, 1998; Arenas et al., 2015), especially in areas with extensive crops where the availability of habitat remnants is scarce (Coffin, 2007). Adjacent spaces to these roads are also used by livestock animals (Coffin, 2007), which play an important role in seed dispersal, especially along the infrastructure (Arenas et al., 2017).

Landscape genetics is an emerging discipline devoted to explaining how the genetic variability and the gene flow among populations could be affected by landscape conditions, including habitat loss and fragmentation (Adams & Burg, 2015; Kierepka & Latch, 2015; Suni & Witheley, 2015). It is experiencing exponential growth, exploring patterns in different species (Segelbacher et al., 2010; Manel & Holderegger, 2013).

Gypsum soils are one of the most diverse and threatened habitats of arid and semi-arid climates worldwide (Escudero et al., 2015), giving shelter to a very specialised flora with a high number of endemics, many of them highly endangered and narrowly distributed (Romão & Escudero, 2005; Martínez-Hernández et al., 2011). These habitats are natural island-like scenarios since they are formed of gypsum outcrops immersed in soils of another nature, creating outcrops surrounded by a matrix inhospitable to these specialists (Escudero et al., 2015; Matesanz et al., 2018). Nonetheless, anthropic fragmentation due to agriculture intensification is exacerbating this natural insularity and posing, in many cases, extreme risk to most of these diverse hotspots (Mota et al., 2004; Escudero et al., 2015). A recent work suggested that some of these soil specialists are resilient enough to face this anthropogenic fragmentation, probably due to an evolutionary history in which natural fragmentation was the norm (Matesanz et al., 2017, 2018), therefore they are adapted to fragmentation: high evolutionary potential, phenotypic plasticity, generalist outcrossing, potent soil seed bank, etc (Matesanz et al., 2017, 2018). However, the over-imposition of an additional barrier for dispersers and pollinators could be affecting the gene flow between the habitat remnants, limiting the long-term viability of these soil specialists.

Here, we performed a study of landscape genetics in a widely distributed and abundant Iberian gypsophile, Lepidium subulatum (Eugenio et al., 2012). This is an exceptional plant model for evaluating the effect of linear infrastructure on its gene flow due to several previous studies of the species. It is a common plant in gypsum soils communities, with specific molecular markers available, demographic populations approaches, etc. Previous works clearly indicated this species is flexible enough to guarantee genetic connection between gypsum islands, even in landscapes subject to severe agricultural intensification (Gómez-Fernández, Alcocer & Matesanz, 2016; Matesanz et al., 2017, 2018). Therefore, we could evaluate the effect of the highway on the gene flow without the spatial structure of patches affecting our results. This was done over a substantial area in which remnants of gypsum habitats are dissected by a busy highway. Our working hypothesis is that this barrier may affect genetic flow not only between the two sides of the highway but also along the highway. For this, we used specific genetic markers (Martínez-Nieto et al., 2012) and characterised in spatial terms all the patches in this area. More specifically, we want to answer the following questions: (i) Is there an effect of the highway apparent in the genetic diversity of Lepidium subulatum’ populations? (ii) Are patches isolated according to the distance between them? If so, is the road enhancing the fragmentation or the connectivity between patches? Finally (iii) are genetic parameters, environmental variables and landscape features related and/or modified by the presence of the highway?

Materials and Methods

Study species

Lepidium subulatum L. (Brassicaceae) in an endemic chamaephyte of the Iberian Peninsula with a few populations in North Africa, specialist of gypsum soils (Hernández Bermejo & Clemente, 1993). It is a perennial shrub (up to 26 years; Eugenio et al., 2012), 20–60 cm in height (Palacio & Montserrat-Martí, 2005; Romão & Escudero, 2005). It is a generally entomophilous and primarily self-incompatible species (Matesanz et al., 2015). The species is diploid with eight chromosome pairs (2n = 16; Hernández Bermejo & Clemente, 1993). Outcrossing and atelechorous seed dispersal with seeds covered by a mucilage able to anchor the seeds in the vicinity of mother plants surely affects its ability to connect habitat remnants (Gómez-Fernández, Alcocer & Matesanz, 2016).

Study area

The area is a homogeneous landscape at the south-east corner of Madrid, central Spain. It is characterised by the presence of fragments of gypsum vegetation, surrounded by an intensively managed matrix of dry, extensive crops (coordinates in UTM ETRS89 30N: the centroid of the most north-westerly population (Pop 13) 4,438,624, 489,512; the most south-easterly population (Pop 12) 4,432,335, 495,277). The climate is arid, with an annual average temperature and rainfall of 13.8 °C and 440 mm, respectively and almost no rainfalls during the summer (Meteorological Station of Arganda del Rey, 459,310, 4,462,676; 38–48 km away from the study area). The area is dissected northwest to south east for more than 15 kms by the A-3 highway, which has operated for more 35 years in its present configuration (Madrid-Valencia, a motorway with four traffic lanes and a width of 25 m; Boletín Oficial del Estado, 2016), dividing the area into two similar halves.

We chose 24 populations or patches distributed at both sides of the A-3 motorway, selecting the same number of populations on each side of the area in a strio of 3 km from the highway (Fig. 1) and maximising the range of fragment sizes and connectivity. Populations were selected from the kilometre point 65 to 75 along the highway. In each gypsum patch we collected samples from 8–20 individuals (average 16.2 ± 4.2 individuals) for DNA extraction (Table S1). We calculated L. subulatum population density as the number of intersected individuals in a transect of 250 m across the patch. Remnants with insufficient number of individuals were discarded to avoid confounding effects on flow arising from small population sizes. For each remnant patch, the following variables were also estimated fragment area, patch perimeter and distance to the highway from the patch’s edge and from the centroid using ArcGis 10.8 (ESRI, 2020). In addition, we calculated the connectivity among patches with Tremlová and Münzbergová’s connectivity index (Tremlová & Munzbergová, 2007). This was done taking into consideration all the remnants present in the area (see Arenas et al., 2017 for details).

Figure 1 Situation of populations or patches in the study area.

In blue, the populations situated in the northeast (NE) A-3 highway side are stand for, and in red, the populations situated in the southwest (SW) of the highway. To make the figure, we used ArcGis using the 2015 public map service IGN-PNOA WMS (https://pnoa.ign.es) as a base layer; and it is an open source document (https://www.boe.es/boe/dias/2015/12/26/pdfs/BOE-A-2015-14129.pdf; Page 3, Article 4.1; 2015).

In order to connect genetic parameters with the physical-chemical characterisation of each patch (remnant’s quality), we collected two soils samplings per patch with a core ten centimetres deep and six centimetres in diameter. Samples were analysed for total amount of nitrogen (N), phosphorous (P), potassium (K), organic carbon, pH, conductivity and glucosidase and phosphatase’s activity in Nutrilab/URJC (https://nutrilab-urjc.es/). We subjected the samples to a sulphuric acid digest and then used an SKALAR San++ Analyser to estimate N and P (Skalar, Breda, Nertherland; Maestre & Puche, 2009). For K, samples were shaken with distilled water in the ratio 1:5 prior to digestion (Maestre & Puche, 2009). We used the Walkley–Black method (Gelman, Binstock & Halicz, 2012; Sato et al., 2014) for estimating organic C. We measured pH and conductivity using a solution 1:2.5 (weight:volume). We used the protocols described in Maestre & Puche (2009) to measure glucosidase and phosphatase activity.

DNA extraction, amplification and sequencing

Material collection was approved by the Consejería de Educación e Investigación de la Comunidad de Madrid (REMEDINAL-TE CM, P2018/EMT-4338). The genetic study was performed with eight specific microsatellites markers: Lsub01, Lsub02, Lsub03, Lsub04, Lsub05, Lsub07, Lsub08, Lsub12 (Martínez-Nieto et al., 2012). DNA was extracted from 60 mg of dried young leaf tissues using the SpeedTools Plant Extraction kit (Biotools, Madrid, Spain). We amplified individually each microsatellite using Polymerase Chain Reactions, mixing 14.2 μl of milli-Q water, 2 μl of T10X buffer with MgCl2, 0.8 μl of dNTP mix [10 mM], 0.5 pmol of each primer (forward labelled with fluorescence dye and reverse) 1 U of Taq-polymerase (Biotools, Madrid, Spain) and 1.2 μl of DNA (DNA had a concentration range of 10–30 ng/ul). The thermocycler’s protocol was: 4 min at 95 °C, followed by 30 cycles of 45 s at 95 °C, 45 s at 51 °C (annealing temperature) and 1 min at 72 °C, followed by a final step of 7 min at 72 °C. PCRs were checked using electrophoresis in agarose gel pre-stained with Gel Red (Biotium, Hayward, CA, USA). Lastly, the amplified fragments were analysed with an ABI 3730XL sequencer (Applied Biosystems, Foster City, CA, USA) of the Unidad de Genómica (Universidad Complutense, Madrid, Spain).

Statistical analysis

We tested the Hardy-Weinberg equilibrium with Genepop 4.7 (Raymond & Rousset, 1995; Rousset, 2008) and the presence of null alleles by MicroChecker 2.2.3 (Van Oosterhour et al., 2004). We also calculated the allelic richness per population/patch, the total number of private alleles, the expected and observed heterozygosity (HE and HO), the inbreeding coefficient (F), and the number of migratory individuals (Nm), using GenAlEx 6.5 software (Peakall & Smouse, 2006). Complementarily, migration rates or number of migrants were estimated with BayesAss 3.04 (Wilson & Rannala, 2003) and Migrate 4.4.3 (Beerli, 2009), to estimate recent or past gene flow rates respectively. We calculated the FST with a null alleles correction with FreeNA (Chapuis & Estoup, 2007; Chapuis et al., 2008). Due to the different numbers of individuals sampled in each population, the allelic richness was calculated by rarefaction analysis using HP-Rare v 6 June 2006 programs (Kalinowski, 2005). To evaluate the existence of any genetic bottleneck, we used the Wilcoxon test in Bottleneck v.1.2.02 software (Cornuet & Luikart, 1996a). This measures deviations in heterozygosity and deficiencies in allelic richness. It takes into consideration that, when a bottleneck is produced, the effective population size drops, triggering a decrease in the allelic number that exceeds the effects of heterozygosity (Cornuet & Luikart, 1996b). We run three different mutation models: infinite allele model (I.A.M.), step wise-mutation model (S.M.M.) and intermediate two-phase mutation model (T.P.M.). We used the parameters by default for T.P.M. (70% of proportion of S.M.M. and 30% of variance). For each model, we performed 2,000 iterations. We also evaluated whether any of the genetic parameters (i.e., allelic richness, private alleles, HO, HE, F, FST average or Nm average) were affected by some soil variables (i.e., N, P, K organic carbon, pH, conductivity, phosphatase and glucosidase) which are surrogates of habitat quality, connectivity (Ci), other landscape variables (fragment size, perimeter of the remnant and minimum distance to highway) and population density. For this, we utilised General Linear Models (GLMs) using R v.3.3.0 (R Core Team, 2016); in which we tested a model with all independent variables but without interactions among predictors. P-values were corrected using Bonferroni’s adjust to avoid the multiple comparison effect.

We studied the genetic structure of the 24 populations using a Bayesian clustering method with STRUCTURE v.2.3.4 with prior information on population membership. Ten independent runs were carried out for each K value (i.e., number of potential clusters, ranging from 1 to 27), each one with a burn-in period of 105 iterations followed by 106 MCMC iterations and 104 thinning. Analyses were developed with allelic correlation frequencies and genetic admixture. According to the different population sampled, we used the STRUCTURESelector (Li & Liu, 2018) module to obtain K values. We considered the Evanno method (Evanno, Regnaut & Goudet, 2005), together with MedMeaK (median of means), MedMedK (median of medians), MaxMeaK (maximum of means) and MaxMedK (Maximum of medians) proposed by Puechmaille (2016) with a threshold value = 0.6, to discard spurious clusters and detect potential substructure. Also, to complement the STRUCTURE analysis (Puechmaille, 2016; Janes et al., 2017), we performed a spatial Principal Components Analysys by adegenet v. 2.2.1 package in R (Jombart, 2008; Jombart & Ahmed, 2011) to evaluate the global and local spatial structure.

To evaluate the effect of the highway as a modifier agent of the genetic landscape, genetic diversity and differentiation was estimated, considering the two sides of the motorway (NE vs. SW population groups) separately. In addition, an AMOVA (Analysis of MOlecular VAriance), performed by GenAlEx 6.5 software (Peakall & Smouse, 2006), was used to estimate and evaluate the distribution of the percentage of intra- and interpopulation variability, including both highway sides as different regions. These results also complemented the genetic structure analyses (see below). To test if the highway triggered a barrier effect between both sides, we calculated the average of FST of one side (northeast or southwest) with each population of the other highway side to observe if the value was significantly different with the FST average of the first side. We also analysed the effect of highway and geographic distance between pairs of populations using Generalised Least Squares (GLS) models with the nlme and corMLPE packages (Pinheiro et al., 2018; Pope, 2019). The corMLPE package constructs regressions using distance matrices with non-linearity and random effects. In this case, we selected two populations and used as independent variables the side of highway (binary variable: 0 = same side of the motorway; 1 = different side) and the distance between populations. We selected the best model using AIC, using FST as the dependent variable and geographic distance, highway side and their interactions as the independent variable.

Results

Patch genetic characterisation

None of populations in remnants were in H-W equilibrium, except population 13 and 14 (Table S1). Some microsatellites could present some null alleles (Table S1). We found an average allelic richness per patch/population of 7.3 ± 1.3 alleles. Population 21 presented the highest value (n = 10.4 ± 1.8) and population 4, the lowest (n = 4.6 ± 1.2) (Table 1; Table S1). When the average number of alleles was rarefied, the average richness decreased to 4.9 ± 1.1 (Table 1). In this case, population 4 continued as the poorest (n = 3.8 ± 0.9), whereas population 12 was the richest (n = 8.3 ± 0.6) (Table 1; Table S1). The total number of private alleles was 35, but several populations did not have any (populations 1, 8, 12, 15, 16, 17, 18, 19, 20) (Table 1; Table S1). In contrast to allelic richness, population 4 had the highest number of private alleles (n = 5) (Table S1). Expected heterozygosity was higher than observed in all populations (average HE = 0.7588 ± 0.0 and average HO = 0.6507 ± 0.1, respectively). Population 21 showed the highest expected heterozygosity, while population 4 had the lowest (HE = 0.8±0.0 and HE = 0.7±0.1, respectively) (Table S1). Nevertheless, the population with the highest observed heterozygosity was population 5; and the lowest, population 12 (HO = 0.8 ± 0.2 and HO = 0.5 ± 0.3, respectively) (Table S1). Finally, the average inbreeding coefficient (F) was 0.1 ± 0.1, with great variation between populations. Population 5 had the lowest inbreeding coefficient, showing a patent heterozygosity excess (F = 0.0 ± 0.2); meanwhile, population 12 had the highest coefficient (F = 0.3 ± 0.3) (Table S1). The presence of bottlenecks was detected for all populations. (Table S3).

Table 1 Average values of genetic parameters of the study area and the both sides of the A-3 highway.

		Total of populations	NE side	SW side	
Allelic Richness	Average	7.282	8.115	7.542	
SD	1.267	1.037	1.450	
Rarefied Allelic Richness	Average	4.914	4.751	5.078	
SD	1.070	0.212	1.514	
Private Alleles	Total	35	13	22	
HE	Average	0.758	0.784	0.732	
SD	0.049	0.025	0.054	
HO	Average	0.650	0.673	0.626	
SD	0.073	0.054	0.082	
F	Average	0.135	0.135	0.135	
SD	0.085	0.077	0.095	
Nm	Average	3.993	5.088	2.800	
SD	1.180	0.910	0.850	
Note:

Genetic parameters were allelic richness, private alleles number, expected and observed heterozygosity (HE and HO, respectively), inbreeding coefficient (F) and migratory individuals number (Nm).

Genetic differentiation and migrants: highway effect

The average number of migratory individuals between population/patches (Nm) was 4.02 (Table 1). Population 22 had the highest number of migratory individuals (Nm = 5.6); and population 12, the lowest (Nm = 1.4) (Table S1). Genetic differentiation based on FST coefficients, showed an average value between population pairs of 0.1, suggesting moderate genetic differentiation across the study area (Table S1) suggesting certain degree of isolation.

When comparing the populations located on either side of the highway (see Fig. 1), there were significant differences in the number of private alleles and number of migratory individuals. In the case of private alleles, the populations of the SW side presented a higher number (n = 22 and n = 13, respectively) (Table 1), while the NE half had a higher number of migratory individuals (Nm = 5.1and Nm = 2.8, respectively). The NE population with the highest number of migratory individuals was population 22 and the population with the lowest was population 24 (Nm = 6.6 and Nm = 3.2, respectively). In contrast, in the SW side, the population with the highest number of migratory individuals was population 1 whereas the lowest number was found in population 12 (Nm = 3.8 and Nm = 1.2, respectively). A similar pattern was found in the evaluation of recent gene flow (i.e., BayesAss). All migration rates were smaller than one migrant per generation, although higher rates were found in populations at both sides of the highway (i.e., population 15 at NE side and population 7 at SW side), forming small clusters (Table S4). Ancient gene flow showed a similar pattern (i.e., Migrate), with populations with high and small migration values at both sides of the highway (Table S5). Only two populations (pop 15 and 3) showed high values of gene flow with both approaches.

The best GLS model or explaining differences in FST values contained as predictors the Euclidean distance between patches or the highway side (Table 2). Therefore, FST was higher when geographic distance among populations was higher or when populations belonged to different highway side. However, these differences should be considered with caution, due to the small reduction in AIC values when this last variable was included in the model (Table 2).

Table 2 General least square models with AIC information values and ΔAIC.

Model	AIC	ΔAIC	
Null	−1471.159	0	
FST ~ geographic distance	−1469.273	−1.886	
FST ~ highway side	−1469.203	−1.956	
FST ~ geographic distance + highway side	−1467.317	−3.842	
FST ~ geographic distance × highway side	−1465.485	−5.674	

Genetic structure and association with environmental variables

Bayesian analyses suggested different number of genetic clusters (K). Namely, the Evanno method suggest K = 4 as the most plausible value. MedMedK and MedMeaK support a K = 7 and MaxMedK and MaxMeaK indcated that K = 8 (Fig. 2). These differences suggest the existence of some internal substructure within the main 4 clusters. This heterogeneity in the number of clusters is the result of internal divisions suggesting the existence of nested structures. For instance, some of the clusters proposed by K = 4 split into different clusters with K = 7 or K = 8. The cluster that groups populations 16–24 are divided into three different clusters in K = 7 or 8 (populations 17,18 and 19 in one cluster, population 24 in by itself, and the other populations in another cluster). In contrast, sPCA showed a global structure (p-value = 0.01) but no a fine scale pattern (p-value > 0.05; Figs. 3C and 3D). This was surveyed in the map of genetic clines (Fig. 3A), where individual scores were similar in all the study area, except in populations 3, 15–19 and 11–12, and in the eigenvalues (Fig. 3B), where only it is remarked one among the positive values (global structure). In the case of the linear models, we found that some genetic descriptors, such as allelic richness (estimate = 0.001; p-value < 0.01) and expected heterozygosis (estimate = 0.01; p-value = 0.04), were positively related to the population density of the patch (Fig. 4). We did not find any other relation with the rest of genetic parameters and environmental variables.

Figure 2 Genetic structure in the study area.

Probability of an individual belongs to a cluster when (A) K = 2, K = 4, K = 7 and K = 8 obtained by structure; and (B) values of Delta K in which K = 4 is the most probable structure.

Figure 3 Spatial Principal Components Analysis.

(A) sPCA surface with the individual scores, population network (closest neighbor was 20) in black lines and the highway in grey line. Axes are in WGS84 coodinate system; (B) the eigenvalues of sPCA; (C) histogram of simulated values for the global structure and the observed value (black point and line); (D) histogram of simulated values for the local structure and the observed value (black point and line).

Figure 4 Relationship between (A) allelic richness and population density and (B) expected heterozygosity and population density.

Regression lines and p-value were obtained using GLMs.

Discussion

The fragmentation paradigm has been mainly constructed by surveying some demographic and/or fitness estimates of individuals, populations or communities (Fahrig, 2003). Unfortunately, fragmentation effects remain uncharacterised for most habitats and, moreover, its consequences (both negative and positive) for many species remain unknown (Fahrig, 2003, 2019). This emerging paradigm needs the explicit consideration of the so-called genetic perspective to include not only current ecological processes but also evolution, combining fragmentation with other factors that could shape the genetic population structure (Aguilar et al., 2008; Matesanz et al., 2017). In this context, linear infrastructure has emerged as powerful fragmentation driver, inducing negative effects on genetic diversity and fluxes (Balkenhol & Waits, 2009; Holderegger & Di Giulio, 2010). Nonetheless, most of studies have focused on animals, especially from a genetic perspective (Balkenhol & Waits, 2009; Holderegger & Di Giulio, 2010). In this context, we evaluated the effects of fragmentation and the barrier effect of a highway in a gypsum soil plant specialist, Lepidium subulatum, considering not only the landscape structure but also other indicators of ecological quality of the habitat remnants together with some population features. It have been emphasised that soil specialists could have reduced evolutionary potential because specialisation could constitute an evolutionary dead end if environmental conditions shift (Anacker et al., 2011; Bieger, Rajakaruna & Harrison, 2014). Nonetheless, recent evidences showed that population genetics of some gypsum specialists suggested a significant resilience and plasticity to adverse conditions (Martínez-Nieto et al., 2012; García-Fernández et al., 2018; Matesanz et al., 2018; Cohen, 2019). In this sense, Lepidium subulatum, which shows an outstanding ecological and genetic variability at different spatio-temporal scales, is a valuable plant model to study not only fragmentation but also its response to other global change drivers (Gómez-Fernández, Alcocer & Matesanz, 2016). Even more, this species showed minimal local adaptation, being its populations’ phenotypic differentiation and local adjustment amounted to phenotypic plasticity (Matesanz et al., 2020a).

Path genetic differentiation

Populations of L. subulatum presented significant variation in genetic diversity, gene flow and structure (e.g., allelic richness and heterozygosity). This variation might be associated with selective drivers that constrain the populations, but also to neutral processes that may foster different levels of genetic diversity in the remnant populations, including biological, demographic and ecological features. Although selective and neutral processes are acting, L. subulatum should sustain levels of genetic diversity that are enough to maintain plastic responses to different environmental conditions (Matesanz et al., 2020a, 2020b). L subulatum is a generalist entomophilous and primarily self-incompatible species (Loveless & Hamrick, 1984; Gómez-Fernández, Alcocer & Matesanz, 2016), with an important and long-standing presence in the soil seed bank (Caballero et al., 2005) and significant plant recruitment under field conditions (Escudero et al., 2000). These factors may support the relatively high levels of genetic diversity, even with the presence of selective forces and demographic or stochastic events which could tend to minimise this variation.

We detected evidences suggesting recent bottlenecks or founder events. Perhaps, due to these, some patches presented a particular concern due to the existence of local moderate F inbreeding coefficients. This probably is due to crossing with close relatives within patches (dispersal is very inefficient, see Escudero et al., 2000), causing an increase of kinship and, therefore, inbreeding (Fischer, Hock & Paschke, 2003). It is well known that inbreeding (or high kinship similarity) has a negative effect on population dynamics because it reduces the reproductive success of the species (Jolivet, Rogge & Degen, 2013; Hermansen et al., 2015) and the frequency of heterozygosity, fixing more easily any deleterious alleles (Mattey & Smiseth, 2015) which potentially decrease the adaptive ability of the species (Porcher & Lande, 2016). However, as the loci number is low, these F inbreeding coefficient values could be due to presence of null alleles.

Genetic differentiation and migrants: highway barrier

Our most remarkable result is the weak effects of the highway on the genetic structure (allelic richness, heterozygosity, genetic differentiation, gene flow parameters) of this species. This type of infrastructure could act as a barrier but could also improve the gene flow between the both sides due to the movement of vehicles or animals, with principal or auxiliary roads acting as dispersal pathways (Schmidt, 1989; Tikka, Högmander & Koski, 2001). Other potential causes that might explain the reduced effect of the highway could be related to the relatively small scale of the study (along 10 km of the motorway), the molecular tool employed (i.e., neutral markers, Balkenhol, Waits & Dezanni, 2009) and the fact that it would be necessary additional generations to verify any genomic signal in the populations (see Eugenio et al., 2012). It is worthwhile to note that some populations on the SW side of the study site might be suffering a potential barrier effect, as showed by the relatively high FST values between populations pairs (Table 1; Table S1). Therefore, it would be interesting to perform a detailed study to separate the possible existence of genetic differentiation derived from selection or demographic dynamics that could suggest potential local extinction risks.

Remnant populations retained most of the genetic diversity of the species in the area, which has been demonstrated as essential to maintain the capabilities of the species to cope with the uniqueness of this habitat (Matesanz et al., 2020a). The presence of certain levels of gene flow (present and ancient) and the moderate FST values could indicate that there is not a deficit in heterozygotes in the remnants. The exchange of genetic material could be mainly favoured by some pollination features (Aguilar et al., 2008; Matesanz et al., 2015; Gómez-Fernández, Alcocer & Matesanz, 2016) and by the cattle herds and their movements. The existence of dense soil seed banks (Eugenio et al., 2012) can also ameliorate the genetic differentiation among populations. On the other hand, in some patchy populations, there is a high number of private alleles. This genetic parameter is related to the amount of genetic flow among patches (Slatkin & Barton, 1989), and could indicate a reduction of genetic flow among populations, or at least, suggests some particular population dynamics in those patches. The number of private alleles is higher than those obtained in the other study of the species (Gómez-Fernández, Alcocer & Matesanz, 2016). This discrepancy could be due to differences in the study areas that favour or restrict genetic flow, for instance, the shape or the size of the gypsum habitat remnants.

Genetic structure and association with environmental variables

Estimates of the genetic structure differed slightly depending on the followed approach. We obtained K = 4 when using Bayesian analyses (i.e., STRUCTURE), while 20 neighbours were necessary to connect the network in the sPCA, although the global structure was also the most important. Nonetheless, both cases showed an important admixture of population of both sides of the highway. Consequently, the highway did not exert a significant barrier effect. This high level of admixture in the genetic clustering might be associated with the presence of different factors that foster the dispersal of the species and maintain the high levels of genetic diversity. Efficient seed dispersal among patches could be associated with the active presence of cattle or mainly stationary (or partially stationary) sheep herds within the area. Highways could work as a barrier but also improve plant dispersal, since vertebrates used roads to move, creating corridors (Suárez-Esteban, Delibes & Frediani, 2013). In this sense, several studies have showed the role that herbivores plays in seed dispersal in gypsum specialists (Pueyo et al., 2008). Sheep seem to have a remarkable effect in the short and long-distance seed dispersal (Manzano & Malo, 2006; García-Fernández et al., 2019), even when seeds do not have the appropriate surface structures (e.g., seed hooks; Bakker et al., 1996; Kuiters & Huiskes, 2010). Foraging dynamics could profoundly affect the structure of the whole landscape since sheep can graze remnants in an unpredictable pattern and occurs variably as a function of the season and palatable biomass.

Surprisingly, we found only marginally significant relationships between genetic parameters and the environmental or landscape variables. As it occurs with other edaphic specialists, L. subulatum only grows in gypsum soils and such specialisation probably acts as an almost binomial filter for the species; the species is excluded of non-gypsum soils but is very abundant in any adequate gypsum island (Palacio et al., 2007; Escudero et al., 2015). Nevertheless, only population density had a positive effect in the allelic richness and the expected heterozygosis, suggesting that patches with a higher number of individuals, independently of other soil or landscape conditions, show higher levels of genetic diversity. This, in accordance with controlled studies in common garden experiments, induces high levels of phenotypic plasticity, a necessity for facing stressful conditions (Matesanz et al., 2020a).

Conclusions

Fragmentation effects in this edaphic specialist seems to be negligible. Concurrent effect of a busy highway with agriculture-induced fragmentation appears insufficient to affect its genetic structure and viability. As described in other fragmented gypsum scenarios, biological features, demographic dynamics and population structure of these species seem a pre-requisite to allow them to evolve as soil specialists in edaphic island environments and to maintain their competitiveness with other species in such harsh conditions. Man-made fragmentation and exacerbation of this island-like landscapes do not seem a problem for the viability of the species. Phenotypic plasticity, together with a generalist pollination, an efficient dispersal mechanism combining short, dense accumulation of seeds in the vicinity of mothers and occasional long-distance dispersal by livestock, guarantee its persistence.

Supplemental Information

Supplemental Information 1 Average values of genetic paramenters.

Presence of null alleles, Hardy-Weinberg equilibrium, average values of allelic richness, private alleles’ number, expected heterozygosity (HE), observed heterozygosity (HO), inbreeding coefficient (F), genetic differentiation (FST) and migratory individuals’ number (Nm). The highway side informs about the populations’ situation respect to the highway.

Click here for additional data file.

Supplemental Information 2 Pairwise FST obtained with FreeNA between populations and population average values obtained to test the possible barrier effect of the highway.

Click here for additional data file.

Supplemental Information 3 The obtained p-values after testing bottlenecks using the Bottleneck software.

Click here for additional data file.

Supplemental Information 4 Number of migrant individual per generation obtained with BayeAss.

The upper part of the matrix shows the gene flow that a population emits. The lower part od the matrix shows the gene flow that a population receives.

Click here for additional data file.

Supplemental Information 5 Values obtained with Migrate.

The upper part of the matrix shows the gene flow that a population emits. The lower part od the matrix shows the gene flow that a population receives.

Click here for additional data file.

We are indebted to Carlos Diaz (URJC) for his support during field work and to Iñaki Mola for all his help during the ECONECT project.

Additional Information and Declarations

Competing Interests

Author Contributions

Field Study Permissions

Data Availability

The authors declare that they have no competing interests.

Irene Martín-Rodríguez performed the experiments, analyzed the data, prepared figures and/or tables, authored or reviewed drafts of the paper, and approved the final draft.

Adrián Escudero conceived and designed the experiments, analyzed the data, authored or reviewed drafts of the paper, funding, and approved the final draft.

Alfredo García-Fernández conceived and designed the experiments, performed the experiments, analyzed the data, prepared figures and/or tables, authored or reviewed drafts of the paper, and approved the final draft.

The following information was supplied relating to field study approvals (i.e., approving body and any reference numbers):

Material collection was approved by the Consejería de Educación e Investigación de la Comunidad de Madrid (REMEDINAL-TE CM, P2018/EMT-4338).

The following information was supplied regarding data availability:

Data is available at Figshare:

Martín-Rodríguez, Irene; Escudero, Adrián; García-Fernández, Alfredo (2020): Limited effect of a highway barrier on the genetic structure of a gypsum soil specialist. figshare. Dataset. DOI 10.6084/m9.figshare.13135568.v1.

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
