# Peer review of "Limited effect of a highway barrier on the genetic structure of a gypsum soil specialist"

_PeerJ, doi:10.7717/peerj.10533_

## Round 0.1 · original submission · Major Revisions

The reviewers and I agree that there is promising material here but that substantial revision of this manuscript is needed. Both reveiwers have provided detailed feedback and this should help you develop the manuscript further.

·

Basic reporting

In my opinion there is no need for revision of the English (but also not my mother tongue). The manuscript provided sufficient field background with good article structure. On the other hand, several figures and table are still a lot of room for improvement.

Experimental design

I believe that the questions investigated here are very relevant to understand plant edaphic species adaptation and conservation in general. Thereby, the study is of broad interest for PeerJ readership. Although analysis methods described with sufficient detailed and information, I fee that additional analyses need for this study.

Validity of the findings

The conclusions are speculative because of weak evidence based on only 7 (or 6) loci. Furthermore, several interpretations for the results of analyses were wrong.

Additional comments

Martín-Rodríguez and colleagues investigated the genetic status of a soil endemic Lepidium species occurring in the fragmented landscape due to a linear infrastructure (i.e., highway). Using analyses of nuclear microsatellite as well as statistical analyses, the authors evaluated genetic diversity, structure, gene flow, the effects of landscape factors. The authors found a high level of genetic diversity and complex spatial genetic structure in the species with the presence of frequent gene flow among patches. They concluded that fragmentation effects due to agriculture and infrastructure seem to be negligible for this species and do not limit species viability.
I believe that the questions investigated here are very relevant to understand plant edaphic species adaptation and conservation in general. The study is of broad interest for PeerJ readership. I think one main challenge to its publication of this study is clarity – as is, I had a hard time following many aspects of the manuscript. I am particularly concerned with the “gene flow” and “genetic structure” results section for which I was not able to interpret the same findings as the authors. The authors have limited the estimation of gene flow among populations as well as the genetic spatial structure. In my opinion, simulation-based gene flow inferences can provide valuable insights into the barrier effects of highway. I would like to see more robust analyses here of contemporary and historical migration. Perhaps use BAYESASS and MIGRATE.
Overall, although I feel that the aim of this study is interesting, there is still a lot of room for improvement. In my opinion there is no need for revision of the English (but also not my mother tongue).


Detailed Comments:
Abstract
• Background is too long. On the other hand, Methods and Results are shallow. It should be more concretely.
• Line 54-55: IBD was not fully validated in this study. I believe that even GLM would not suffice.
• Line 56-57: Population’s genetic diversity and structure is usually the result of biological, demographic and ecological features. Thus, no need to mention again.
• Line 57-58: I feel that the interpretation is biased. Is the genetic diversity is an only indicator for displaying the barrier effects of highway?
• Line 58-59: I cannot follow the logic.

Introduction
• Line 71-31: This sentence is irrelevant to the conclusion and aims of this study.
• Line 111-113: This sentence is ambiguous. I think this explanation (i.e., evolutionary history) is too brief so it will be hard for readers to understand why soil specialists are resilient against to fragmentation due to human activities. Please be more specific.
• Line 114: It may lack the term “gene” or “pollen” before “flow”.
• Line 117-124: The former (L117-121) and later (L. 123-124) are disconnected logically. Previous studies should have been indicated that the species exhibits high levels of genetic connectivity between the fragmented patches. Is this understanding of mine correct? If correct, these previous works suggest that there is no barrier preventing gene flow and/or migration in this area. So I cannot fully understand the argument for your hypothesis.

Materials and methods
• Line 135-143: Is there any information available about the generation time of the species? It may be important for this study.
• Line 145: “... corner of Madrid (central Spain).”, better “... corner of Madrid, central Spain.”
• Line 145-153: The position information (i.e., latitude and longitude) concerning “the area” should be provided in the text or figure 1.
• Line 179-181: Only seven marker names were listed in the text.
• Line 189-191: “... the samples were amplified with an ABI 3730XL” is wrong. “the amplified fragments were analyzed with an ABI 3730XL” is better.
• Line 203-206: Parameters for TPM model should be exhibited.
• Line 211-213: Please provide package and function name used here.
• Line 225-227: Why did you use DAPC among several model-free methods? I recommend to substitute a spatially explicit multivariate method principal component analysis (sPCA, Jombart et al. 2008) for DAPC. This can allow to test fine scale genetic structure, and to estimate the strength of spatial structure.

Results
• Overall, STRUCTURE analyses would not be affected by null alleles, but other analyses might. I would suggest doing a null-allele correction on FST.
• Line 247-253: The statistics related to number of alleles (e.g., allelic richness) should be described as one digit after the decimal point (also in Table1 and S1).
• Line 254-256: “Expected heterozygosity was higher than...”, better “A relatively high level of heterozygosity observed in all populations”.
• Line 263-263: “discarded” is wrong, “detected” is correct.
• Line 268-270: I think that the level of genetic differentiation is low. And also, readers may want to the values of pairwise FST between populations but not within each population.
• Line 296: You should provide the delta K value or the log likelihood values for each K.
• Line 299-303: Several results discussed here were not presented graphically so it will be hard to understand. It can be informative to visualize lower and upper hierarchical genetic structures for relevant K (e.g. K = 3, 7, 8) in addition to the main observed genetic structure (K 4).
• Line 303-305: This sentence should move to Discussion section.
• Figure 2 (A): “Geneland” is not explained in Materials and Methods section. Is this a result of STRUCTURE analysis? If so, I wonder that all individuals within a population were assigned by single cluster despite the low genetic differentiation and frequent migrants were observed in the studied populations.
• Figure 2 (C): Are there reasons for order of populations in the figure? If there is not it, populations should be rearranged in coordinate longitude or population code order.

Discussion
• Line 310-341: The two paragraphs are redundant. It can be shortened.
• Line 358-360: Your interpretation about the Bottleneck test is wrong. I think that the results of Bottleneck suggest the presence of recent bottlenecks in the all populations. Cornuet and Luikart (1996) said that “If P-value is lower than 0.05, the null hypothesis is rejected in favor of the hypothesis of an overall heterozygosity excess and a recent genetic bottleneck” (Genetics 144, pp. 2005).
• Line 359-360: I guess so, but you should pay more attention to a high variance among loci. The interpretations are speculative because of weak evidence based on only 7 (or 6) loci. The high FIS within populations may be attributable to the presence of null alleles.
• Line 369: Which genetic parameter suggests the weak effects of highway?
• Line 375-377: Why do you think that only SW population is suffering in a barrier effect? There is no anthropogenic barrier within the SW side (population). I believe that FST values were not clearly different between both side.
• Line 381-383: A complex spatial genetic structure does not always mean a high level of genetic diversity.
• Line 394-396: A distance from highway may be candidate to explain high private allele, if the highway functions as an effective gene flow barrier.
• Line 404-406: OK. When moving, do these livestock use the highway (or side road)? If they use, the high way may play important role in seed flow among populations.

Reviewer 2 ·

Basic reporting

The article title “Limited effect of a highway barrier on the genetic structure of a gypsum soil specialist” addresses an engaging question on the effect of fragmentation on the genetic structure of a plant. I consider that the manuscript is in general well structured, with some points to improve. The authors show a broad knowledge in the study system, but they might improve the descritpion of their study species (e.g. seed dispersal distances, longevity,…). The proposed hypothesis is interesting and well justified and they have designed a proper methodological framework to test it. I miss some interesting points in the Discussion (e.g. the role of time in delaying the impact of fragmentation on genetic structure) as I specify below. These are my specific comments:

Abstract (line 50-51). “Here, the effect of "a" highway with heavy traffic on the genetic flow and diversity of Lepidiuma dominant shrubby gypsophyte, was evaluated.” I think that this sentence needs to be rephrased.

Abstract (line 55). “….exploring in detail the effect of "a" highway….”

Introduction (line 67-73). The first paragraph is very important. Readers need to see the importance of the issue that will be studied in the manuscript and how your research is going to shed light in the field. You start the intro talking about landscape genetics, this is the methodological framework you are going to use to answer your question, but this is not the ecological problem you are addressing. Your manuscript is about fragmentation, so the first sentence must talk about how fragmentation threaten biodiversity worldwide. Then, you can talk about how landscape genetics approaches are suitable to address these kind of questions and how your study will contribute to increase the knowledge in the field. Please change the structure of this paragraph.

Introduction (line 86). “…profoundly affects the "gene" flow between…”

Introduction (line 102). Change dispersion for dispersal.

Introduction (line 117-120). You say that your plant is a perfect model species to study the effect of linear infrastructure because its known capability of maintaining gene flow in highly fragmented landscapes. I do not see that as a positive point for your study. If you select a species which tolerates high levels of fragmentation, you will not probably detect any effect of this perturbation and you are probably overlooking the effect of this perturbance in other species of this habitat which have less ability to cope with fragmentation. I think that you must change this part. You can say that your species has the ability of maintaining gene flow, but this does not justify that your species is a good model for this study. Maybe you might say that it is a good model because there is a lot of previous information, there are molecular markers available, it is a widespread species in the community, etc.

Introduction (line 124). “…sides of the highway but also among the fragments in each side”. I do not understand how your experimental design research on the effect of the highway among the fragments in each side, what is the effect of the highway in the connectivity among fragments in each side?

Study species (line 134). What do we know on the role of animals in seed dispersal of the species? Could you include information on this issue? As is covered by a mucilage, might it be dispersed by birds?


Study area (line 157-159). You say that you collected 20 individuals where possible. I could not find the number of individuals collected in each population in any of the tables or files, could you please provide this information?

Study area (line 161-165). How did you estimate fragment area, patch perimeter….? I suppose that you used a GIS tool. Please include this information.

DNA extraction, amplification and sequencing. Did you amplify the microsatellites in individual pcr or in multiplex? Please add this information.

Statistical analysis. You did not calculate the frequency of null alleles. This is an important parameter to verify that your genetic estimates are accurate. You have to check that. I think it could be also interesting to check if your populations are in Hardy-Weinberg equilibrium as this parameter might impact some of your results (e.g. STRUCTURE).

Statistical analysis (line 221-227). You apply 2 clustering methods. Please justify that. You might read this reference for this part: Janes, J. K., Miller, J. M., Dupuis, J. R., Malenfant, R. M., Gorrell, J. C., Cullingham, C. I., & Andrew, R. L. (2017). The K= 2 conundrum. Molecular Ecology, 26(14), 3594– 3602.

Statistical analysis (line 230). Which software did you use to perform the AMOVA? Please specify that.

Results (line 305-307). I suppose that you did not find any relationship among genetic descriptors and the remaining environmental variables, but please say it.

Discussion (general comment). I think that time can be someting important to consider in your Discussion. This could be a critical question for your study. Fragmentation can need time to be detectable. It is quite common to find a delay in the impact of fragmentation on genetic descriptors. How long can the individuals of your plant live? Is there any information about how long can seeds remain in the soil seed bank? The highway was built 35 years ago, maybe this is not enough time how to see genetic effect in your species. Please consider to include this aspect in your Discussion.

Discussion (general comment). Is there any information on seed dispersal distances or pollen dispersal distances in this species? It might be something important to comment

Discussion (line 404). Change dispersion for dispersal

Conclusion (line 427). Eliminate "has been". The final sentece "As described....."

Figure 1. How did you elaborate the map? Please say it

Figure 2. You used STRUCTURE not GENELAND, right?

Experimental design

The experiemental design is suitable for the proposed question. Authors are cautious about their results since the spatial scale is restricted, but it is an interesting case study to report. Some points of the methology need to be improved (specified in Basic reporting)

Validity of the findings

Conclusions are supported by the results. However, some points of the methodology can be improved to strengthen the accuracy of the results

---

## Round 0.2 · Minor Revisions

There remain a few minor corrections to make after which the manuscript can be accepted. Thank you for your work in adressing these issues raised in the first review.

·

Basic reporting

included the general comments.

Experimental design

included the general comments.

Validity of the findings

included the general comments.

Additional comments

This is my second time reviewing this manuscript, and I found that the authors have done a good job in addressing the reviewer’s comments. I have no further comments, but I picked on several minor things:
• Abstract: It is kind for readers to include DNA maker information, such as the number of loci and type (i.e., nuclear or chloroplast).
• Line 149: A period is need in these geographic coordinates (e.g., 44.38624).
• Line 213: Again, parameters for T.P.M. model should be exhibited.
• Line 224: STRUCTURE’s “K” is better in Italic.
• Line 225: Why authors conducted STRUCTURE analysis ranging from K = 27? I think K = 1 to 24 (= the number of focal populations) is enough for this study.
• Line 300: “Meanwhile”, better “Namely”
• Line 301-303: This sentence is too long and complex, so should be simply.
• Line 318-342: Also, this paragraph is long, so divide two paragraphs.
• Figure 2: For readers, the name of population group (i.e., NE or SW) should be in the bar plots. In addition, in the delta K methods, K = 2 was relatively high. I’m not only one who wants to see the results of K = 2 assignment.
• Figure 3: The labels on these axes are too small and hard to see.
• Figure 4: “p” is better in Italic.
• “All” supplemental files: Very hard to see without a caption on the tables. Such irresponsible jobs can lead to doubts about the quality of this paper. That was unfortunate.

Reviewer 2 ·

Basic reporting

Well-written.

Enough background information.

Relevant results.

Experimental design

Good design.

Methods are suitable to address the proposed hypotheses.

Validity of the findings

Findings are valuable. Conclusions are relevant in the field. However, the spatial scale is quite small, and hence, it is difficult to do general assumptions with these results.

Anyway, a valuable case study.

Additional comments

The manuscript has been improved. Authors applied most of the suggested changes and I consider that the manuscript can be now accepted.

I only have two suggestions. It is a little bit strange to me the way that you report null alleles in Table1. From my point of view, you shoul report a frequency by marker and/or population. Maybe you have null alleles but the frequency is low, this can not be evaluated in the current version.

You said that you collected 20 individuals where possible. However, you only got 20 individuals in 7 populations. You should give a range (between 8 and 20 individuals, average XXX+-XXX). This would be more informative.

---

## Round 0.3 · accepted · Accept

There remain a few corrections to the English to make:

Short title should be: Highway effects over gypsophyte genetics (delete 's' from gypsophyte)

Line 41 evolutive should be evolutionary

Lines 44-46 Here, we evaluated the effect on the genetic flow and diversity of a highway with heavy traffic in the species Lepidium subulatum, a dominant Iberian shrubby gypsophyte. Should read - Here, we evaluated the effect of a highway with heavy traffic on the genetic flow and diversity in the species Lepidium subulatum, a dominant Iberian shrubby gypsophyte.

Line 188 needs the concentration of the dNTP mix

Line 190 needs the concentration range of the DNA

Thank you for making the corrections requested.